# In Vitro Physico-Chemical Characterization and Standardized In Vivo Evaluation of Biocompatibility of a New Synthetic Membrane for Guided Bone Regeneration

**DOI:** 10.3390/ma12071186

**Published:** 2019-04-11

**Authors:** Lívia da Costa Pereira, Carlos Fernando de Almeida Barros Mourão, Adriana Terezinha Neves Novellino Alves, Rodrigo Figueiredo de Brito Resende, Marcelo José Pinheiro Guedes de Uzeda, José Mauro Granjeiro, Rafael Seabra Louro, Mônica Diuana Calasans-Maia

**Affiliations:** 1Graduate Program in Dentistry, Universidade Federal Fluminense, Niterói, Rio de Janeiro 24.020-140, Brazil; livia_costa3@hotmail.com; 2Oral Surgery Department, Dentistry School, Universidade Federal Fluminense, Niterói, Rio de Janeiro 24.020-140, Brazil; mouraocf@gmail.com (C.F.d.A.B.M.); resende.r@hotmail.com (R.F.d.B.R.); mjuzeda@oi.com.br (M.J.P.G.d.U.); dr.rafaelseabra@gmail.com (R.S.L.); 3Oral Diagnosis Department, Dentistry School, Universidade Federal Fluminense, Niterói, Rio de Janeiro 24.020-140, Brazil; adrianaterezinha@globo.com; 4Bioengineering Program, National Institute of Metrology, Quality and Technology, Duque de Caxias, Rio de Janeiro 24.020-140, Brizil; jmgranjeiro@gmail.com

**Keywords:** membranes, polymer, biocompatibility, inflammatory cells, calcium phosphate

## Abstract

This study’s aim was to evaluate the biocompatibility and bioabsorption of a new membrane for guided bone regeneration (*polylactic-co-glycolic acid* associated with hydroxyapatite and β-*tricalcium phosphate*) with three thicknesses (200, 500, and 700 µm) implanted in mice subcutaneously. Scanning electron microscopy, X-ray diffraction, Fourier transform infrared spectroscopy, and the quantification of carbon, hydrogen and nitrogen were used to characterize the physico-chemical properties. One hundred Balb-C mice were divided into 5 experimental groups: Group 1—Sham (without implantation); Group 2—200 μm; Group 3—500 μm; Group 4—700 μm; and Group 5—Pratix^®^. Each group was subdivided into four experimental periods (7, 30, 60 and 90 days). Samples were collected and processed for histological and histomorphometrical evaluation. The membranes showed no moderate or severe tissue reactions during the experimental periods studied. The 500-μm membrane showed no tissue reaction during any experimental period. The 200-μm membrane began to exhibit fragmentation after 30 days, while the 500-μm and 700-µm membranes began fragmentation at 90 days. All membranes studied were biocompatible and the 500 µm membrane showed the best results for absorption and tissue reaction, indicating its potential for clinical guided bone regeneration.

## 1. Introduction

The bone is a specialized connective tissue capable of regenerating completely. Therefore, regenerative procedures are used to gain bone tissue, allowing professionals the possibility of rehabilitating a patient both aesthetically and functionally. However, bone tissue’s growth pattern requires a longer time for formation compared to non-mineralized connective tissue, since fibroblasts proliferate and produce matrices of collagen more quickly than do the osteoblasts that form bone tissue [1].

Guided Bone Regeneration treatment (GBR), in which the regeneration of defects is predictably achieved by the application of occlusive membranes, was developed based on the Guided Tissue Regeneration (GTR) concept. GBR involves mechanically excluding non-osteogenic cells from the surrounding soft tissues and allowing osteogenic cells originating from native bone to inhabit the defect, thus preventing the growth of unwanted soft tissue in the bone defect [2]. Experimental studies have provided significant evidence that bone regeneration is significantly increased by mechanically preventing the invasion of soft tissues into bone defects [3,4].

In the last 40 years, different membranes have been developed to act as physical barriers in the treatment of bone defects. Today, resorbable collagen and aliphatic polyester membranes are the most commonly used materials to eliminate the need for a second surgery for membrane removal [5,6]. However, collagen undergoes rapid degradation due to the action of inflammatory cells in the region, which results in poor mechanical resistance that leads to collapse [7].

Among the polymers used, the most applicable in medicine are those that, when degraded, generate soluble and non-toxic products that are compatible with the organism. Aliphatic polyesters derived from lactic acid and glycolic acid undergo hydrolytic degradation. However, despite the excellent mechanical properties of polylactic acid (PLA), its biocompatibility, biodegradability, hydrophobicity, brittleness, acidic degradation products and high cost restrict its applications [8]. The PLA polymer has a slower hydrolysis rate compared to the polyglycolic acid (PGA) polymer in the human body, so for a suitable degradation of the polymer, the PLA polymer has been combined with the PGA polymer as a copolymer, thereby improving its characteristics [9]. 

*Polylactic-co-glycolic acid* (PLGA) alone is not an osteoconductive material, so associating calcium phosphates with this polymer improves the interaction between the polymer and the precursor cells of new tissue. In this way, osteoconductivity can be introduced into non-biological materials, such as metallic, ceramic and synthetic polymers, through various strategies, including coating and the incorporation of osteoconductive materials to form composites, such as the incorporation of calcium phosphate in its structure [10,11,12,13]. Inserting calcium phosphate-hydroxyapatite (HA) and β-*tricalcium phosphate* (β-TCP) prevents collapses by improving mechanical properties [14]. 

GBR may be influenced by the time required and the tissue reaction caused by reabsorption of the membrane components. The X-Ray diffraction quantification (XRD) method, Fourier transform infrared spectroscopy (FTIR), scanning electron microscopy (SEM) and the quantification of carbon, hydrogen and nitrogen allow investigating the structural composition and surface characteristics of the membrane, evaluating its mineral phases, measuring the crystallinity and quantifying the constituents and surface porosity.

Accordingly, this study involved using mice as an experimental model due to their rapid prolificity, easy handling, known biology and genome and inexpensive maintenance, as well as similarity of mice and human results. The aim of this study was to subcutaneously evaluate biocompatibility and bioabsorption according to the criteria established by ISO 10993-6: 2016.

## 2. Materials and Methods

### 2.1. Control Membrane

Commercially available *polylactic-co-glycolic acid* (PLGA) dental membrane (Pratix^®^, Baumer, São Paulo, SP, Brazil) was purchased for comparing the biocompatibility and bioabsorption rates in the subcutaneous tissue of rats. 

### 2.2. Physico-Chemical Characterization of Experimental Membranes

The crystalline mineral phases present in the membranes, their crystallinity and the proportion of the HA and β-TCP e phases were examined by XRD. The XRD patterns were obtained using an Empyrean-Panalytical diffractometer (Almelo, Netherlands) operating at 45 kV and 40 mA, with CuKα radiation (Cu—1.540598 Å), a temperature of 25 °C and relative air humidity of about 55%. The data were collected in the 2θ range of 20–60° with a step of 0.02° points per second. 

The contents of the HA and β-TCP phases in the samples were evaluated by the relative intensities of specific peaks of β-TCP and HA XRD patterns in the sample, as described by Balmain et al. [15]. The vibrational modes of phosphate and hydroxyl groups in samples were analyzed using FTIR. The spectra were obtained with a Thermo Scientific, Nicolet iS50 (Madison, WI, USA) operating in transmission mode from 650 to 4000 cm^−1^: Scans number—8; Resolution—4 cm^−1^.

The microstructure of the membrane was investigated by using scanning electron microscopy (SEM) (FEG-ZEISS^®^-mod. SUPRA 55VP, Oberkochen, Germany) of the surface and the transversal section at 300, 1000, 3000, 10,000 and 20,000×. 

For SEM, the samples were mounted on a stub of metal with adhesive, coated with 40–60 nm of a metal such as Gold/Palladium and then observed in the microscope. 

The SEM images were obtained using an Axio Imager m2m-Zeiss microscope (Göttingen, Germany). The membranes were cut into 1-cm pieces and placed in a holder at 90° positions. The obtained images were increased by 20×. Image analysis was performed using Axiovision SE64 software 4.9.1 (Göttingen, Germany). The carbon, hydrogen, and nitrogen contents were quantified in duplicate using the organic elemental analyzer.

For histological preparation, the samples were fixed in 10% formalin solution for a minimum of 48 h at room temperature, then the samples were dehydrated through a series of graded ethanol baths to displace water and infiltrated with wax. The infiltrated tissues were then embedded into wax blocks cut into 5 µm pieces and stained with Hematoxylin and Eosin. 

### 2.3. Animal Characterization and Experimental Group

This study was carried out in compliance with the guidelines of the 3Rs Program (Reduction, Refinement and Replacement), whose objective is to reduce the number of animals used during experimentation, to minimize pain and discomfort and avoid euthanasia at the end of experimentation (NC3Rs 2010); the experiments were reported according to the ARRIVE guidelines regarding relevant items. The Ethical Committee of the Universidade Federal Fluminense approved the study and the protocol no. is CEUA/UFF: 869. One hundred Balb-C mice, male and female, weighing approximately 30 g, were provided by the Laboratory Animals Center at Fluminense Federal University (Niterói, Rio de Janeiro, Brazil). The animals were divided into 5 experimental groups: Group 1—Sham (without membrane implantation); Group 2—PLGA membrane + HA + β-TCP (200 μm); Group 3—PLGA membrane + HA + β-TCP (500 μm); Group 4—PLGA membrane + HA + β-TCP (700 μm); and Group 5—Pratix^®^ membrane implantation. The materials were supplied by FGM Materiais Odontológicos LTDA (Joinville, Santa Catarina, Brazil). All experimental groups were subdivided into 4 experimental periods (7, 30, 60 and 90 days) with 5 animals in each group/experimental period. Before and after the study, all animals were kept in isolators with a maximum of 5 animals in each and fed with food pellets and water ad libitum.

### 2.4. Surgical Procedure and Production of Samples

After a 24-h fast, all animals were submitted to general anesthesia by the intraperitoneal route, following Fluminense Federal University protocol, with a 0.6-mL injection of anesthetic solution prepared with 1.0 mL of 10% Ketamine (Dopalen^®^-100 mg/mL), 0.5 mL of 2% xylazine (Anasedan^®^ 20 mg/mL) and 8.5 mL of sterile saline (KabiPac^®^). 

Three minutes later, degermation and trichotomy were performed with chlorhexidine degermant and chlorhexidine alcoholic 2% solutions (Riohex Scrub^®^, Rioquimica; São José do Rio Preto, São Paulo, Brazil). An approximately 1-cm-long incision was made in the epithelium of the animal’s dorsal region, followed by divulsion of the muscular fascia skin with the aid of scalpel and blunt-tipped scissors to expose the subcutaneous tissue for insertion of the membrane (1 cm) into the subcutaneous region. This was followed by a 5-0 nylon suture (Ethicon^®^, Johnson & Johnson, São Paulo, Brazil) and antisepsis with gauze and alcoholic chlorhexidine solution (Figure 1).

In the postoperative period, the animals were kept in the Animal Experimentation Laboratory (AEL/UFF) and divided in isolaters based on their experimental groups, where they received food and water ad libitum. On the day of surgery and on each of the following two days, 5 mg/kg of Meloxicam (Eurofarma Laboratórios LTDA, São Paulo, SP, Brazil) was administered subcutaneously every 24 h.

The mice were euthanized after the respective experimental periods with lethal doses of anesthetic solution and the samples and adjacent subcutaneous tissue (±5 mm with safety margins) were collected, fixed, decalcified, dehydrated, clarified and included in paraffin to obtain 5-μm thick slices. The slices were stained with Hematoxylin and Eosin (HE) and observed using a light field microscope at 40× magnification (Nikon Eclipse E400, Tokyo, Japan). These images were captured using a high-resolution digital camera (Sony^®^ HD DSC HX9V 16.2 Mega Pixels, Tokyo, Japan) with a 10× acroplan objective lens at the Laboratory of Applied Biotechnology—UFF (LABA, Niteroi, Brazil) for descriptive and semiquantitative histological evaluation for the presence of inflammatory infiltrate, vascular neoformation, extension and type of necrosis, presence of fatty infiltrate and fibrosis and membrane bioabsorption.

## 3. Results

### 3.1. Structure and Characteristics of the PLGA Membrane

Figure 2 shows the synthetic membrane of PLGA. The membranes exhibited a micro-fibrous appearance that indicates guided bone regeneration, as demonstrated in Sanaei-Rad et al.’s study [16] in which this fibrous architecture mimics the extra-cellular matrix and was shown to be suitable for cell attachment, proliferation and differentiation. The micro-fibrous surface exhibited a highly porous structure with interlaced non-woven fibers. The diameters of the fibers ranged from 0.2 to 2 μm. The micro-fibrous appearance can be seen in the different thicknesses of membranes in Figure 2.

The XRD evaluation showed two crystalline mineral phases present in the membranes (HA and β-TCP), as well as the proportions of HA (54.2%) and β-TCP phases (45.8%) (Figure 3). Table 1 shows the mean values for Carbon (39.4%), Hydrogen (4.5%) and Nitrogen (0.1%). The carbon, hydrogen and nitrogen contents of the membrane with 500 µm were quantified in duplicate using the organic elemental analyzer.

When analyzing the FTIR spectra of the membrane, the PLGA polymer and the calcium phosphates used for its preparation, we observed no peaks above 3500 cm^−1^ on the membrane, which could indicate degradation of the polymer by hydrolysis, according to Tan et al. [17]. The membrane spectrum corresponds to a junction of the characteristic peaks of BTCP and HAP (peaks identical in FTIR) with the characteristic peaks of PLGA (Figure 4). 

### 3.2. In Vivo Implantation

Overall, the animals tolerated well the anesthesia, pre-surgery and post-surgery, without complications or setbacks. Biological effects after the implantation of the different experimental membranes were evaluated according to the criteria established by ISO 10993-6: 2016/Part 6/Annex E and the descriptive analysis of the tissue response to the membranes was evaluated as a function of tissue disposition in the different membranes: bioabsorption, the presence of inflammatory cells, vascular neoformation and the presence of fibrosis. The degree of inflammation was evaluated and quantified manually according to the number and distribution of inflammatory cells present at the membrane-tissue interface: that is, polymorphonuclear cells, lymphocytes, plasma cells, macrophages and giant cells. In addition, the degree of degeneration (debris) was determined by morphological alterations due to necrosis extension.

For the 7-day experimental period, small vascular neoformation with minimal capillary proliferation (presence of 1 to 3 bulbs per examined field) was observed in the Sham Group. Also found in the Sham Group were a cicatricial process with an absence of multinucleated giant cells and macrophages and an abundance of lymphocytes and plasmacytes, in addition to the moderate presence of polymorphonuclear cells. In Group 2 (200 μm), a moderate infiltration of lymphocytes, macrophages and multinucleated giant cells was observed, along with a slight presence of polymorphonuclear cells. Moderate vascular neoformation with capillary proliferation was also present (presence of a wide range of capillaries per examined field). 

In Group 3 (500 μm), a moderate infiltration of lymphocytes and macrophages and a discrete infiltration of plasmacytes and multinucleated giant cells were observed, alongside mild vascular neoformation with a proliferation of capillaries. Group 4 (700 μm) presented integral membranes with a moderate presence of lymphocytes, plasm cells and macrophages and a discrete polymorphonuclear infiltration. Local neovascularization with small capillary proliferation was also observed in this group. The control Group (Pratix ^®^) did not show the membrane in any cuts performed; therefore, the cellular response tissue of the periphery membrane was evaluated. There was a moderate presence of lymphocytes, macrophages and polymorphonuclear infiltration. Also observed in this group was moderate local neovascularization with small capillary proliferation (Figure 5).

In the 30-day experimental period, a discrete presence of lymphocytes and plasm cells and a moderate presence of macrophages were observed in the Sham Group. Regarding tissue response, a discrete presence of vascular neoformation with few vascular proliferations characterized this sample. In Group 2 (200 μm), membranes began to exhibit fragmentation with tissue invasion. There was a moderate presence of multinucleated giant cells and macrophages, a discrete presence of lymphocytes with an absence of polymorphonuclear cells and mild local neovascularization.

The membranes of the 500-μm and 700-μm Groups showed no changes in integrity after 30 days. Group 3 showed a discrete presence of lymphocytes, plasma cells, macrophages and a few multinucleated giant cells, with a moderate tissue response of local neovascularization. Group 4 had a discrete presence of lymphocytes and plasm cells, a moderate presence of multinucleated giant cells and few polymorphonuclear cells. Regarding tissue response, there was moderate vascular neoformation with proliferation of more than 7 shoots per field. The Pratix^®^ membranes used as a control group did not adhere to the subcutaneous tissue, so during removal of the samples they were detached from the tissue and the tissue adjacent to the implantation site of this membrane was observed, which revealed discrete lymphoplasmacytic infiltration and a moderate infiltration of macrophages (Figure 6).

After 60 days, the discrete presence of lymphocytes, plasm cells and macrophages was observed in the Sham Group. There were no multinucleated giant cells or polymorphonuclear cells and there was a slight presence of vascular neoformation shoots. In Group 2, there was a moderate presence of lymphocytes, macrophages and multinucleated giant cells, a discrete presence of plasm cells and an absence of polymorphonuclear cells. Groups 3 and 4 presented a discrete/moderate presence of lymphocytes, macrophages, multinucleated giant cells and plasm cells. However, while Group 3 presented mild vascular neoformation, Groups 2, 4 and 5 showed moderate neovascularization. Group 5 was macroscopically detached from the tissue and unconnected to the subcutaneous tissue. The presence of multinucleated and polymorphonuclear giant cells was not observed and there was only a moderate presence of lymphocytes and macrophages (Figure 7).

After the 90-day period, Group 1 showed discrete lymphocyte and macrophages infiltration and an absence of multinucleated giant cells and polymorphonuclear cells. The 200-μm group showed a moderate presence of macrophages, lymphocytes and multinucleated giant cells and a discrete presence of plasma cells. In the 500-μm and 700-μm groups, there was no presence of polymorphonuclear cells or plasmocytes and a moderate presence of lymphocytes, multinucleated giant cells and macrophages. The control group had a discrete presence of polymorphonuclear cells and a moderate presence of macrophages, multinucleated giant cells, lymphocytes and plasma cells. All presented moderate vascular neoformation, with the exception of Group 1, which had discrete neoformation (Figure 8). 

There was no presence of necrosis, fibrosis or fatty infiltration for all experimental periods.

According to ISO, at the end of the evaluation, test samples may present scores ranging from 0.0 to 2.9 (absence of tissue reaction), 3.0 to 8.9 (discrete tissue reaction), 9.0 to 15 (moderate tissue reaction) and above 15.1 (severe tissue reaction). Thus, after evaluation of each cell type and tissue within the conditions of this study, following ISO 10993-6/2016, it was observed that the 200-μm test sample showed a slight tissue reaction (4) compared to the control sample after 7 days of implantation, whereas the 500-μm (2.6) and 700-μm (0) test samples showed no tissue reactions after 7 days of deployment. After 30 days, the tested 700-μm membrane showed a slight tissue reaction (4.4) when compared to the control sample, whereas the 200-μm (1.2) and 500-μm (1.8) samples did not present tissue reactions. After 60 days, the 200-μm membrane presented a slight tissue reaction (7) when compared to the control group. There was an absence of tissue reactions for the 500-μm (2.6) and 700-μm (3.2) Membranes. After 90 days of implantation, the 700-μm membrane showed tissue reaction (3.8), while the 200-μm membrane obtained a score of 0.6 and the 500-μm membrane obtained a score of 0, characterizing an absence of tissue reaction.

## 4. Discussion

For membranes to be used effectively in the GTR process, they must have biocompatibility characteristics, occlusive properties, the capacity to maintain space, tissue integration and easy clinical manageability [18,19]. The desirable characteristics of those membrane barriers used for GBR therapy include biocompatibility to allow integration into host tissues without creating an inflammatory response, a degradation profile accompanying tissue neoformation, mechanical and physical properties sufficient to allow membrane installation and sufficient sustained force to not collapse and to perform the proper barrier role [20].

Studies have demonstrated that membranes containing PLGA + HA and β-TCP contribute mainly due to their mechanical properties, which improve structural integrity and flexibility [21,22]. In addition, the release of calcium and phosphorus ions during the degradation of HA and β-TCP may be involved in bone metabolism and promote the formation of new bone [23]. Sanaei-Rad et al. [16] demonstrated that HA improves osteogenic properties, strength and structural stability. Seyedjafari et al. [24] evaluated the incorporation of nanohydroxyapatite on the surface of an electro-spun poly (l-lactide)-PLLA associated with human-cord-blood-derived unrestricted somatic stem cells, in vitro under osteogenic induction and *in vivo* after subcutaneous implantation, by demonstrating adequate mechanical properties and improving the osteogenic differentiation of somatic stem cells.

The tested membranes did not present moderate or severe tissue reactions in the experimental periods studied and thus did not differ from the patterns presented by the control group. Therefore, they are biocompatible. The present results demonstrate that all membranes were intact after 7 days of implantation, but after 30 days, the 200-μm membrane was partially fragmented and had lower mechanical stability, while the 500-μm and 700-μm membranes started the process of fragmentation only after 90 days of implantation. The 200-μm membrane showed slight tissue reactions after 7 and 60 days, while the 700-μm membrane showed slight tissue reactions after 30 and 90 days. The 500-μm membrane did not show tissue reactions in any experimental periods. To play its role as a barrier, absorbable membranes should remain for at least three to four weeks [25].

Therefore, we verified that all thicknesses of membranes are biocompatible with the subcutaneous connective tissue of mice; however, the 200-μm membrane presented faster absorption when compared to those of other thicknesses and the control group. The 500-μm membrane only started fragmentation after 90 days, an ideal time to exclude mechanically non-osteogenic and non-tissue response cell populations, thus making it a good thickness for this type of membrane.

This way, we could verify that 3 different versions of the tested membranes were well tolerated by the tissues, but only the 500-μm and 700-μm membranes were intact for the necessary time to allow the GBR process to occur, since the period of degradation could accompany the time required for bone neoformation. Therefore, a clinical study involving an in vivo model should be performed to prove its clinical efficacy.

Particles of thermally deproteinized inorganic bone (hydroxyapatite) have been demonstrated to induce a chronic inflammatory reaction in the presence of giant cells and fibrosis of the particles in the subcutaneous tissue of rats [26]. Therefore, it is reasonable to suggest that the recruitment of giant cells may be related to persisting remnants of the mineral phase on the membrane. 

## 5. Conclusions

Despite the limitations of this preclinical study, it is possible to conclude that the tested membranes are well tolerated by the tissues and are not completely resorbed at 90 days. Therefore, due to the evaluated biocompatibility and bioabsorption of the absorbable membranes derived from PLGA + HA + β-TCP, these membranes could be used as barriers to promote tissue regeneration in surgical techniques of GBR. However, further in vivo (bone repair model) and clinical studies should be conducted to determine the clinical efficacy of these membranes.

## Figures and Tables

**Figure 1 materials-12-01186-f001:**
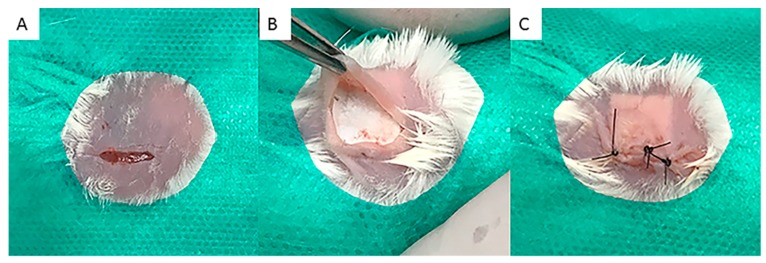
Sequence of surgical procedures performed. (**A**) Trichotomy, skin antisepsis of the back region of the animals and incision of approximately 1cm on the back region of the animals, according to ISO 10993-6/2016; (**B**) After divulsion of the cutaneous tissue, a 1-cm fragment of membrane was implanted and tested according to the experimental groups; (**C**) Replacement of cutaneous tissue over the implanted membrane and suture with 5-0 mononylon (Ethicon^®^, Johnson & Johnson, Brazil).

**Figure 2 materials-12-01186-f002:**
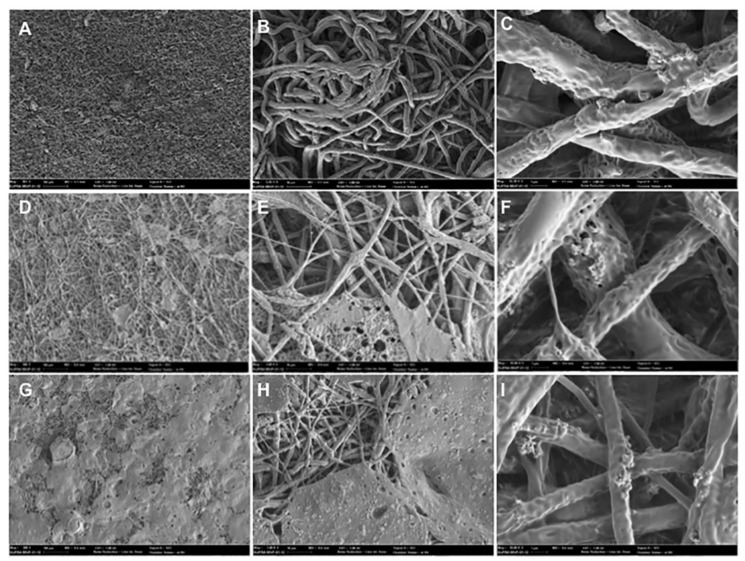
Scanning electron microscopy (SEM) micrographs of the membrane surface at 300× magnification, scale bar = 100 μm (**A**,**D**,**G**); 3000× magnification, scale bar = 10 μm (**B**,**E**,**H**); 20,000× magnification, scale bar = 1 µm (**C**,**F**,**I**). **A**–**C** (200 µm membrane), **D**–**F** (500 µm membrane), and **G**–**I** (700 µm membrane).

**Figure 3 materials-12-01186-f003:**
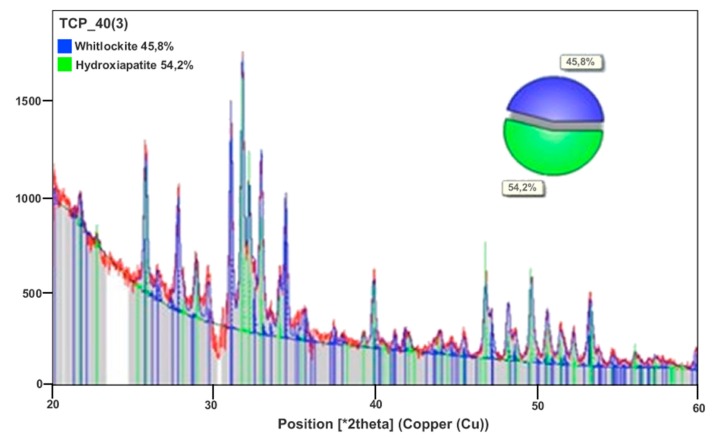
X-ray diffraction pattern (XRD) of the membrane sample. Observe the peaks representing hydroxyapatite and β-TCP. The quantitative analysis showed the presence of two phases: 45.8% β-TCP and 54.2% HA.

**Figure 4 materials-12-01186-f004:**
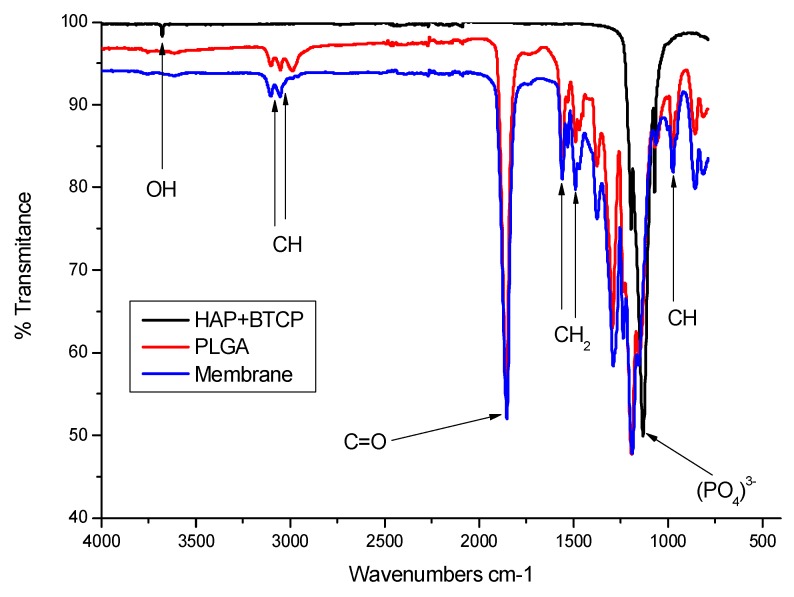
Fourier transformed infrared (FTIR) spectrum of the membrane sample. Observe the bands of OH and (PO_4_)^3−^ characteristics of hydroxyapatite and CH, C=O, CH_2_ which are the main peaks of PLGA; in the blue color graph it is observed that the mixture between PLGA and HA was efficient since all reference bands are evidenced.

**Figure 5 materials-12-01186-f005:**
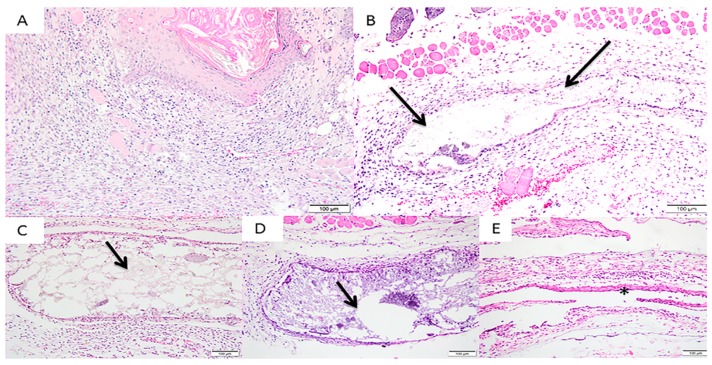
Photomicrographs of the experimental groups after 7 days of membrane implantation in the connective tissue. (**A**) Group 1 (Sham): moderate presence of polymorphonuclear cells and absence of multinucleated giant cells and macrophages; (**B**) Group 2 (200 μm): moderate infiltration of lymphocytes, macrophages and multinucleated giant cells limiting the implanted area; (**C**) Group 3 (500 μm): moderate infiltration of lymphocytes and macrophages and a discrete infiltration of plasm cells and multinucleated giant cells; (**D**) Group 4 (700 μm): moderate presence of lymphocytes, plasm cells and macrophages and a discrete polymorphonuclear infiltration; (**E**) Group 5 (Pratix^®^): moderate presence of lymphocytes, macrophages and polymorphonuclear infiltration. Observe the integrity and the presence of fragment-free membranes in all groups. Arrow corresponds to the implanted membrane; Asterisk corresponds to the Pratix^®^ membrane site. (Magnification: 200×, Stain: Hematoxylin and Eosin).

**Figure 6 materials-12-01186-f006:**
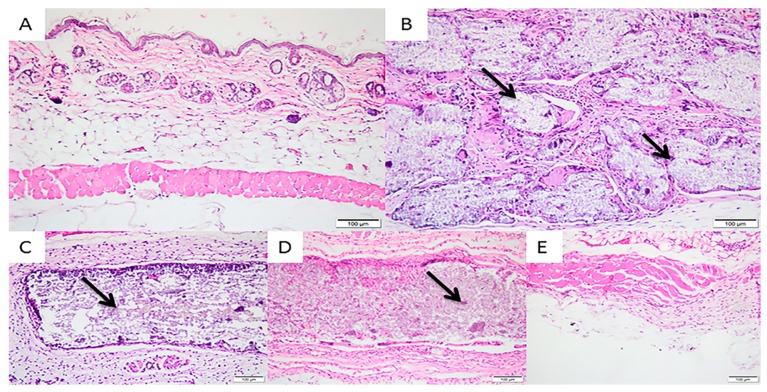
Photomicrographs of the experimental groups after 30 days of membrane implantation in the connective tissue. (**A**) Group 1 (Sham): discrete presence of lymphocytes and plasm cells and a moderate presence of macrophages; (**B**) Group 2 (200 μm): moderate presence of multinucleated giant cells and macrophages and a discrete presence of lymphocytes; (**C**) Group 3 (500 μm): discrete presence of lymphocytes, plasma cells, macrophages and a few multinucleated giant cells; (**D**) Group 4 (700 μm): discrete presence of lymphocytes and plasm cells, moderate presence of multinucleated giant cells and few polymorphonuclear cells; (**E**) Group 5 (Pratix^®^): discrete lymphoplasmacytic infiltration and a moderate infiltration of macrophages. Observe the fragmented aspect of the 200-μm membrane and non-fragmented aspects in the 500-μm and 700-μm groups. Arrow corresponds to the implanted membrane; Asterisk corresponds to the Pratix^®^ membrane site. (Magnification: 200×, Stain: Hematoxylin and Eosin).

**Figure 7 materials-12-01186-f007:**
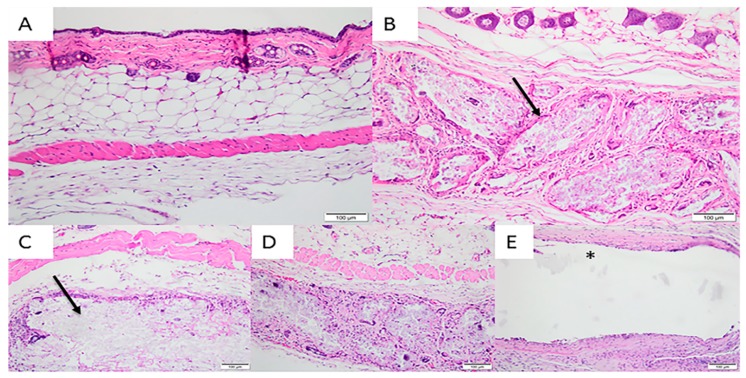
Photomicrographs of the experimental groups after 60 days of membrane implantation in the connective tissue. (**A**) Group 1 (Sham): discrete presence of lymphocytes, plasm cells and macrophages; (**B**) Group 2 (200 μm): moderate presence of lymphocytes, macrophages and multinucleated giant cells, a discrete presence of plasm cells and an absence of polymorphonuclear cells; (**C**) Group 3 (500 μm) and Group 4 (700 µm): discrete/moderate presence of lymphocytes, macrophages, multinucleated giant cells and plasm cells; (**D**,**E**) Group 5 (Pratix^®^): absence of multinucleated and polymorphonuclear giant cells and only a moderate presence of lymphocytes and macrophages. Arrow corresponds to the implanted membrane; Asterisk corresponds to the Pratix^®^ membrane site. (Magnification: 200×, Stain: Hematoxylin and Eosin).

**Figure 8 materials-12-01186-f008:**
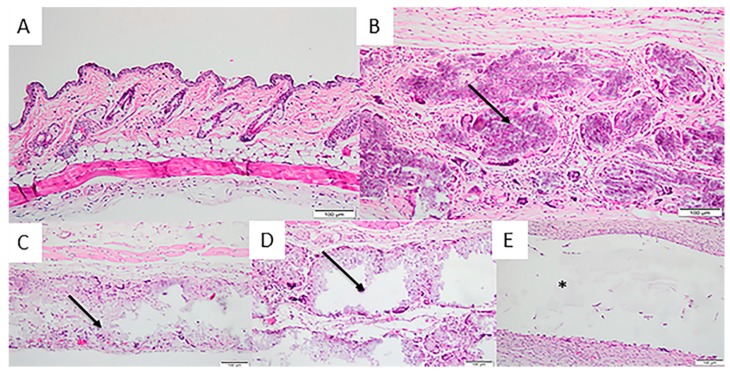
Photomicrographs of the experimental groups after 90 days of membrane implantation in the connective tissue. (**A**) Group 1 (Sham): discrete lymphocyte and macrophages infiltration and an absence of multinucleated giant cells and polymorphonuclear cells; (**B**) Group 2 (200 μm): moderate presence of macrophages, lymphocytes and multinucleated giant cells and a discrete presence of plasm cells; (**C**) Group 3 (500 μm) and (**D**) Group 4 (700 μm): presence of polymorphonuclear cells, plasmocytes and a moderate presence of lymphocytes, multinucleated giant cells and macrophages and (**E**) Group 5 (Pratix^®^): discrete presence of polymorphonuclear cells and a moderate presence of macrophages, multinucleated giant cells, lymphocytes and plasm cells. Arrow corresponds to the implanted membrane; Asterisk corresponds to the Pratix^®^ membrane site. (Magnification: 200×; Stain: Hematoxylin and Eosin).

**Table 1 materials-12-01186-t001:** CHN SO PE 2400 series II, PerkinElmer.

	Carbon %	Hydrogen%	Nitrogen %
1	39.5	4.7	0.1
2	39.2	4.3	0.1
Mean	39.4	4.5	0.1

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
