# Peer review of "In Vitro Physico-Chemical Characterization and Standardized In Vivo Evaluation of Biocompatibility of a New Synthetic Membrane for Guided Bone Regeneration"

_materials, 2019, doi:10.3390/ma12071186_

Round 1

Reviewer 1 Report

The aim of the present study was to evaluate the physico-chemical properties, biocompatibility and bioresorption of newly developed PLGA associated with hydroxyapatite and beta-TCP membranes with three different thicknesses.

The study basically consisted of an in vitro physico-chemical characterization of the three evaluated membranes and of a preliminary in vivo analysis of the same membranes, inserted subcutaneously, in terms of inflammatory response and tissue reaction. The conducted analyses should be better specified in the Introduction as well as in the title; indeed, in the latter the in vitro characterization is not mentioned.  The title should be rephrased as in this format is misleading. 

Moreover, as stated by the Authors in the Conclusions section, the clinical efficacy of the tested membrane in the GBR procedures should be better understood in a in vivo model. Even this aspect should be stressed within the Discussion section.

In addition, some points should be addressed, as reported below:

- Table 1 should not be quoted within Materials and Methods section, but in the Results. Moreover, had carbon, hydrogen and nitrogen contents quantification been performed on the three different membranes? Please, specify within the text and add the results in Table 1.

- Processing of the specimens for SEM analysis as well as for histological evaluation should be described in depth; otherwise Authors should provide literature references supporting the procedures.

- SEM results should be better described; particularly, the following sentence “The membrane exhibited a micro-fibrous aspect to favor guide bone regeneration” should be developed in depth, describing the morphology of the micro-fibrous and how their aspect would favor the GBR. Moreover, the Authors stated that “The difference in the morphology between the different membranes thickness is clearly visible on the 20.000X magnification observed by SEM”; however, the description of the membranes features have not been provided  either in the text or in the Figure 2 caption.

- Figure 5 is missing.

- Captions of figures 6, 7 and 8 should be fully rewritten with more accuracy providing an in depth description of the histological images, and including the meaning of the arrows.

According to this reviewer’s consideration, major revisions of the present manuscript is strongly recommended.

Author Response

Reviewer 1

1. The conducted analyses should be better specified in the Introduction as well as in the title; indeed, in the latter the in vitro characterization is not mentioned.

Answer: The authors have included a new paragraph in the Introduction section that specifies the analyses performed for physico-chemical characterization. This paragraph is highlighted in the revised manuscript. The authors have included the in vitro characterization in the title.

2. The title should be rephrased as in this format is misleading.

Answer: The authors have changed the title and the new title is “In vitro physico-chemical characterization and standardized in vivo evaluation of biocompatibility and bioresorption and of a new synthetic membrane for guided bone regeneration.” 

 3. Moreover, as stated by the Authors in the Conclusions section, the clinical efficacy of the tested membrane in the GBR procedures should be better understood in an in vivo model. Even this aspect should be stressed within the Discussion section.

Answer: The authors have included the following in the Discussion section: “This way, we could verify that 3 different versions of the tested membranes were well tolerated by the tissues, but only the 500-μm and 700-μm membranes were intact for the necessary time to allow the GBR process to occur, since the period of degradation could accompany the time required for bone neoformation. Therefore, a clinical study involving an in vivo model should be performed to prove its clinical efficacy.”

4. Table 1 should not be quoted within Materials and Methods section, but in the Results. Moreover, had carbon, hydrogen and nitrogen contents quantification been performed on the three different membranes? Please, specify within the text and add the results in Table 1.

Answer: The authors have moved Table 1 to the Results section. The quantification was performed in only one membrane (500 µm) and this information was included in the text.

5. Processing of the specimens for SEM analysis as well as for histological evaluation should be described in depth; otherwise Authors should provide literature references supporting the procedures.

Answer: For scanning electron microscopy, the samples were mounted on a stub of metal with adhesive, coated with 40 ‐ 60 nm of a metal such as Gold/Palladium, then observed through the microscope. For histological preparation, the samples were fixed in 10% formalin solution for a minimum of 48 hours at room temperature and the samples were dehydrated through a series of graded ethanol baths to displace the water, then infiltrated with wax. The infiltrated tissues were then embedded into wax blocks cut into 5-µm pieces and stained with Hematoxillin and Eosin. This information is now included in the Materials and Methods section.

6. SEM results should be better described; particularly, the following sentence “The membrane exhibited a micro-fibrous aspect to favor guide bone regeneration” should be developed in depth, describing the morphology of the micro-fibrous and how their aspect would favor the GBR. Moreover, the authors stated that “The difference in the morphology between the different membranes thickness is clearly visible on the 20.000X magnification observed by SEM”; however, the description of the membranes features have not been provided either in the text or in the Figure 2 caption.

Answer: The SEM images were obtained using an Axio Imager m2m - Zeiss microscope. The membranes were cut into 1-cm pieces and placed in a holder at 90° positions. The obtained images were increased by 20x. Image analysis was performed using Axiovision SE64 software. This information is now included in the Results section.

Synthetic polymers provide the flexibility of adapting mechanical properties and degradation kinetics to suit various applications; they may be fabricated in various forms with desired characteristics. The membranes exhibited a micro-fibrous appearance that indicates guided bone regeneration, as demonstrated in Sanaei-Rad et al.’s study [16] in which this fibrous architecture mimics the extra-cellular matrix and was shown to be suitable for cell attachment, proliferation and differentiation. The micro-fibrous surface exhibited a highly porous structure with interlaced non-woven fibers. The diameters of the fibers ranged from 0.2 to 2 μm. The micro-fibrous appearance can be seen in the different thicknesses of membranes in Figure 2.

7. Figure 5 is missing.

Answer: Figure 5 shows the histological photomicrographs after 7 days and it was included.

8. Captions of figures 6, 7 and 8 should be fully rewritten with more accuracy providing an in depth description of the histological images, and including the meaning of the arrows.

Answer: The arrows correspond to the implanted membrane. The captions of Figures 5, 6, 7 and 8 were rewritten and are highlighted in the text.

Reviewer 2 Report

In these paper the authors were evaluated the physico-chemical properties, biocompatibility and bioabsorption of 3 different new membranes for bone guided regeneration (PLGA associated with hydroxyapatite and β-TCP) with three thicknesses (200, 500 and 700 μm) implanted in mice subcutaneously. In my opinion this paper need major modifications related to the following point:

1.       In abstract line 38 the word membrane appears twice.

2.       Figure 2 does not specify the difference between A, D and G or B, E and H or G, H and I.

3.       Figure 4 is not clear, the numbers is too small. Please edit the figure. In the description of the figure, the word of spectra needs to be changed in the spectrum, because the figure contains one spectrum.

4.       Discussion of FT-IR spectrum needed.

5.       Figure 5 is missing.

Author Response

Reviewer 2

1. Figure 2 does not specify the difference between A, D and G or B, E and H or G, H and I.

Answer: The differences between the images are only magnification. In C, F and I, the images show the same pattern of phosphate deposition on the PLGA filaments. These images show rough spots between the fibers.

2. Figure 4 is not clear, the numbers is too small. Please edit the figure. In the description of the figure, the word of spectra needs to be changed in the spectrum, because the figure contains one spectrum.

Answer: The authors have edited and altered the text.

3. Discussion of FT-IR spectrum needed.

Answer: The FTIR of the membrane makes sense if analyzed together with the spectra of the PLGA polymer and the calcium phosphates used to synthesize the membrane. The first spectra corresponding to the phosphate (HAP + BTCP; the FTIR spectrum of the sample shows the characteristic bands for PO3-4 ions (963 cm-1, 1036 cm-1 and 1095 cm-1)). A characteristic feature of the present invention is the ability to adsorb adsorbents with the following characteristics: carbonyl at 1760 cm-1, ester at 1000-1300 cm-1 and carboxyl at around 1650 cm-1. A triple peak is also present at about 1400 cm-1. Lactate-lactate (LL) monomeric units are adsorbed at 1456 cm-1, glycolate glycolate (GG) at 1425 cm-1 and lactate-glycolate (LG) at 1398 cm-1. The spectrum of the membrane is nothing more than a junction of the characteristic peaks of BTCP and HAP (peaks identical in FTIR), with the characteristic peaks of PLGA. Peaks above 87500px-1 were not observed in the membrane, which could indicate degradation of the polymer by hydrolysis. This information has been added to the text. 

4. Figure 5 is missing.

Answer: Figure 5 shows the histological photomicrographs after 7 days and it was included.

Reviewer 3 Report

The authors implant a membrane of PLGA with HA and TCP in its composition, compare it with a commercial membrane of PLGA without HA or TCP. They perform a histological study of the results. I do not see anywhere the effect of HA and TCP on the evolution of the membrane, neither in the discussion part.

In my opinion, this would be the strong point of the paper, to see how the content of HA and TCP affects the evolution of the membrane and the surrounding tissue. The authors do not emphasize this point, only on the degradation of the membrane without relating it to the content of HA and TCP. They do not compare in deep with other authors.

On the other hand, they indicate to do a physico-chemical study of the membrane and there is no physical data of the membranes. Finally, the authors do not indicate at any time how they have obtained the membranes.

Its characterization by Rietveld leaves much to be desired ... ???? as well as incuding a FTIR figure and saying only that there are PLGA peaks when the material also has HA and TCP ….???

Figure 5 is missing

It is a very histological study and little focused on the material (membrane). Material without characterizing properly. Send the paper to a biomaterials /histological journal.

Abstract

The first time an acronym is used, indicate the full name of its meaning. Once indicated the full name followed by the acronym, do not repeat this several times throughout the text, since it is already defined once. Always use the acronym once it has been defined. Please repair all the text.

Do not indicate the elements analyzed, but the technique used for their detection.

Group 1 is only a surgical incision without any type of implant, so I do not understand the meaning of the word Sham.

Indicate the amount of TCP and HA has the membrane.

Line 40 ... Showed the best results ... .. in terms of?

Introduction

Last paragraph. Line 73. "The aim of this study was to evaluate the physico-chemical properties, biocompatibility and ..." the paper did not present any physico-chemical properties: neither % porosity, mechanical stress, mechanical resistance, density, specific surface ...... Please change the phrase or include some physical characterization.

M & M

The membrane of group 5 is commercial. I understand that the membranes studied in this work are not commercial, but obtained in the laboratory, right? In this case, please indicate in m & m how these membranes have been obtained, or failing that, if you already have a publilshed paper, include that reference at the beginning of m & m.

Line 89 XRD range 10-70, but figure 3 at 20-60.

In relation to the study by Rietveld, please indicate the number of the cards used, the database used, the Rietveld program used. Change figure 3 of XRD for a Rietveld XRD, because with the one shown in figure 3 you have not been able to do the refinement by Rietweld. Include the measurement error, eliminate background, include the peaks of the experimental data and those of the HA TCP  ... .. Based on figure 3 and the little said in m & m it is very difficult to determine how they have obtained the value of the 45.8 / 52.2.%

Figure caption 3 indicates XRD of membrane sample. What of all the membranes? Do they all have the same HA and TCP content? How have HA and TCP included in the membrane?

How have the sample been prepared for FTIR?

Do not give results in m & m (table 1).

Why have only carbon hydrogen and nitgrogen measured in the membrane? Would it not be logical to also include phosphorus and calcium content, given that they have HA and TCP?

 3 Results.

Figure 2. To make the label magnification clearer inside the figures, it is not seen, and although the authors indicate scale bar = 10 microns in fig. caption you cannot see the line in the photos.

XRD: see my comments in & M.

Make an in-depth study of FTIR results. Say only that there are PLGA peaks when the material also has HA and TCP ??? ..... It's not true. Indicate to which vibration mode each peak belongs: example P-O-P; PO43-; ... symetrical or asimetrical, streching, bending ... .. Everything is missing!

As figure 5 is missing, the text between line 187 and 210 can not be revised.

In the histology photos increase the mag. Label. It is not appreciated. Also, since the journal is of material, please include in the figures *, or any other signal indicating where the lymphocytes, plasma cells, macrophages and a few multinucleated giant cells are, local neovascularization .... Etc., it is difficult to follow the description of the figures if it is not a histologist.

At the foot of the figure indicate what the arrows mean.

Discussion

What happens with the HA and the TCP of the membrane? How does it influence the evolution of the membrane? Where is calcium and phosphorus?

Does fragmentation of the membrane have anything to do with the content of HA and TCP?

Why quantify the amount of HA and TCP if its influence on the behavior of the membrane is not discussed later?

Where is HA and TCP in the histology photos? Or the neoformed tissue?

A physical chemical study of the membrane is indicated, but there is no data from the physical study; and from the chemical study they analyze nitrogen, carbon and hydrogen, but not calcium and phosphorus, when the membrane has Ca and P.

There is a mineralogical study, but its results are not analyzed in depth. Author talks about other authors like Sanaeri- Rad, but author does not relate it to the authors' membranes. The authors speak of Ca and P ion release but they do not give any data of said release of their membranes.

Author Response

Reviewer 3:

1. The authors implant a membrane of PLGA with HA and TCP in its composition, compare it with a commercial membrane of PLGA without HA or TCP. They perform a histological study of the results. I do not see anywhere the effect of HA and TCP on the evolution of the membrane, neither in the discussion part. In my opinion, this would be the strong point of the paper, to see how the content of HA and TCP affects the evolution of the membrane and the surrounding tissue. The authors do not emphasize this point, only on the degradation of the membrane without relating it to the content of HA and TCP. They do not compare in deep with other authors.

Answer: PLGA alone is not an osteoconductive material, so associating calcium phosphates with this polymer improves the interaction between the polymer and the precursor cells of new tissue. In this way, osteoconductivity can be introduced into non-biological materials, such as metallic, ceramic and synthetic polymers, through various strategies, including coating and the incorporation of osteoconductive materials to form composites, such as the incorporation of calcium phosphate in its structure [10,11,12,13]. This information was included in the Introduction section.

Additionally, inserting calcium phosphate (hydroxyapatite and b-TCP) prevents membrane collapses by improving mechanical properties. This information was added to the Introduction section. HA improved osteogenic properties, strength and structural stability.

Seyedjafari et al. [24] evaluated the incorporation of nanohydroxyapatite on the surface of an electro-spun poly (l-lactide)-PLLA associated with human-cord-blood-derived unrestricted somatic stem cells, in vitro under osteogenic induction and in vivo after subcutaneous implantation, by demonstrating adequate mechanical properties and improving the osteogenic differentiation of somatic stem cells. This information was added to the Discussion section.

Particles of thermally deproteinized inorganic bone (hydroxyapatite) have been demonstrated to induce a chronic inflammatory reaction in the presence of giant cells and fibrosis of the particles in the subcutaneous tissue of rats [26]. Therefore, it is reasonable to suggest that the recruitment of giant cells may be related to persisting remnants of the mineral phase on the membrane. This information is present in the discussion session.

2. On the other hand, they indicate to do a physico-chemical study of the membrane and there is no physical data of the membranes. Finally, the authors do not indicate at any time how they have obtained the membranes.

Answer: The physical data of the membrane was provided from the SEM evaluation. The membranes are not commercially available, so information about the synthesis is confidential.

3. Its characterization by Rietveld leaves much to be desired ... ???? as well as incuding a FTIR figure and saying only that there are PLGA peaks when the material also has HA and TCP ….???

Answer: The FTIR of the membrane makes sense if analyzed together with the spectra of the PLGA polymer and the calcium phosphates used to synthesize the membrane. The first spectra corresponding to the phosphate (HAP + BTCP; the FTIR spectrum of the sample shows the characteristic bands for PO3-4 ions (963 cm-1, 1036 cm-1 and 1095 cm-1)). A characteristic feature of the present invention is the ability to adsorb adsorbents with the following characteristics: carbonyl at 1760 cm-1, ester at 1000-1300 cm-1 and carboxyl at around 1650 cm-1. A triple peak is also present at about 1400 cm-1. Lactate-lactate (LL) monomeric units are adsorbed at 1456 cm-1, glycolate glycolate (GG) at 1425 cm-1 and lactate-glycolate (LG) at 1398 cm-1. The spectrum of the membrane is nothing more than a junction of the characteristic peaks of BTCP and HAP (peaks identical in FTIR), with the characteristic peaks of PLGA. Peaks above 87500px-1 were not observed in the membrane, which could indicate degradation of the polymer by hydrolysis. This information has been added to the text.

4. Figure 5 is missing.

Answer: Figure 5 shows the histological photomicrographs after 7 days and it was included.

5. Abstract - The first time an acronym is used, indicate the full name of its meaning. Once indicated the full name followed by the acronym, do not repeat this several times throughout the text, since it is already defined once. Always use the acronym once it has been defined. Please repair all the text.

Answer: The authors have added the full name of the acronym for the first use.

6. Do not indicate the elements analyzed, but the technique used for their detection.

Answer: The detection was performed with a FLASH 2000 Organic Elemental Analyzer.

7. Group 1 is only a surgical incision without any type of implant, so I do not understand the meaning of the word Sham.

Answer: Sham is the name given to the group without implantation. This group was performed to evaluate the biological response (inflammatory cells) to surgical trauma (incision, detachment and suture).

8. Indicate the amount of TCP and HA has the membrane.

Answer: The proportion is: HA (54.2%) and β-TCP phases (45.8%).

9. Line 40 ... Showed the best results ... .. in terms of?

Answer: All membranes studied were biocompatible, but only the 500-µm membrane showed the best results related to absorption and tissue reaction, showing its potential for bone-guided regeneration.

10. Introduction - Last paragraph. Line 73. "The aim of this study was to evaluate the physico-chemical properties, biocompatibility and ..." the paper did not present any physico-chemical properties: neither % porosity, mechanical stress, mechanical resistance, density, specific surface ...... Please change the phrase or include some physical characterization.

Answer: The authors have changed the sentence to: “Accordingly, this study involved using mice as an experimental model due to their rapid prolificity, easy handling, known biology and genome and inexpensive maintenance, as well as similarity of mice and human results. The aim of this study was to subcutaneously evaluate biocompatibility and bioabsorption according to the criteria established by ISO 10993-6: 2016.”

11. M & M - The membrane of group 5 is commercial. I understand that the membranes studied in this work are not commercial, but obtained in the laboratory, right? In this case, please indicate in m & m how these membranes have been obtained, or failing that, if you already have a published paper, include that reference at the beginning of m & m.

Answer: FGM Dentscare is the manufacturer of the experimental membranes; we don’t have a published paper about those membranes. The authors have included how the membranes were obtained.

12. Line 89 XRD range 10-70, but figure 3 at 20-60.

Answer: The authors have changed the range to 20-60.

13. In relation to the study by Rietveld, please indicate the number of the cards used, the database used, the Rietveld program used. Change figure 3 of XRD for a Rietveld XRD, because with the one shown in figure 3 you have not been able to do the refinement by Rietweld. Include the measurement error, eliminate background, include the peaks of the experimental data and those of the HA TCP  ... .. Based on figure 3 and the little said in m & m it is very difficult to determine how they have obtained the value of the 45.8 / 52.2 %.

14. Figure caption 3 indicates XRD of membrane sample. What of all the membranes? Do they all have the same HA and TCP content? How have HA and TCP included in the membrane?

Answer: The membranes have the same proportions of PAH and BTCP. The method of confecting the membranes is an industry secret.

15. Do not give results in m & m (Table 1).

Answer: Table 1 was removed from MM and added to the results section.

17. Why have only carbon hydrogen and nitgrogen measured in the membrane? Would it not be logical to also include phosphorus and calcium content, given that they have HA and TCP?

Answer: No analysis of the chemical composition of the membrane was carried out, since the polymer eventually degraded during the analysis. FRX analysis was performed to evaluate the chemical composition of calcium phosphates.

18. Results - Figure 2. To make the label magnification clearer inside the figures, it is not seen, and although the authors indicate scale bar = 10 microns in fig. caption you cannot see the line in the photos.

Answer: We have changed the text.

19. Make an in-depth study of FTIR results. Say only that there are PLGA peaks when the material also has HA and TCP ??? ..... It's not true. Indicate to which vibration mode each peak belongs: example P-O-P; PO43-; ... symetrical or asimetrical, streching, bending ... .. Everything is missing!

Answer: Answered according to question 3.

20. As figure 5 is missing, the text between line 187 and 210 can not be revised.

Answer: Figure 5 was included in the text.

21. In the histology photos increase the mag. Label. It is not appreciated. Also, since the journal is of material, please include in the figures *, or any other signal indicating where the lymphocytes, plasma cells, macrophages and a few multinucleated giant cells are, local neovascularization .... Etc., it is difficult to follow the description of the figures if it is not a histologist.

Answer: The authors have included the labels in the figures.

22. At the foot of the figure indicate what the arrows mean.

Answer: The arrows correspond to the membranes.

23. Discussion - What happens with the HA and the TCP of the membrane? How does it influence the evolution of the membrane? Where is calcium and phosphorus?

Answer: They are reabsorbed as expected; however, it is reasonable to suggest that the recruitment of giant cells may be related to the persistence of remnants of the mineral phase on the membrane. As the membrane compositions did not vary in thickness, we can conclude that the difference in the bioabsorption rate was related to the thickness of the material and not in the percentage of HA and BTCP.

24. Does fragmentation of the membrane have anything to do with the content of HA and TCP?

Answer: No, it’s related to the thickness of the material, because the concentration of HA and BTCP was not altered between the different membranes.

25. Why quantify the amount of HA and TCP if its influence on the behavior of the membrane is not discussed later?

Answer: Because the concentration of HA and BTCP was not altered between the different membranes. Rather, the thickness of the membrane was changed.

26. Where is HA and TCP in the histology photos? Or the neoformed tissue?

Answer: See the answer to question 23. They are reabsorbed as expected; however, it is reasonable to suggest that the recruitment of giant cells may be related to the persistence of remnants of the mineral phase on the membrane (HA and BTCP).

27. A physical chemical study of the membrane is indicated, but there is no data from the physical study; and from the chemical study they analyze nitrogen, carbon and hydrogen, but not calcium and phosphorus, when the membrane has Ca and P.

Answer: See the answers to questions 3, 6, 17 and 19.

28. There is a mineralogical study, but its results are not analyzed in depth. Author talks about other authors like Sanaeri- Rad, but author does not relate it to the authors' membranes. The authors speak of Ca and P ion release but they do not give any data of said release of their membranes.

Answer: FRX analysis was performed to evaluate the chemical composition of the calcium phosphates in the membranes. The evaluation of tissue reaction to membrane degradation was made based on histological analysis following ISO 10993-6: 2016; it was observed that all membrane thicknesses were well tolerated by the tissues and no severe tissue reactions were observed according to the classification criteria.

Round 2

Reviewer 2 Report

While the authors have address most of my concerns, and I appreciate the fact that they introduced new FT-IR measurements to get better data, there are still some points which remain confusing and must be addressed prior to acceptance of this manuscript:

1. The FT-IR spectra in present form can not be evaluated, because the bands are not clear, the differences are not well defined. Please interpret the results according to the literature. The X and Y axis titles are missing.

2. Please move the sentences “The SEM images were obtained using an Axio Imager m2m - Zeiss microscope.” and “The obtained images were increased by 20x. Image analysis was performed using Axiovision SE64 software.” in materials and methods section.

Author Response

Dear Reviewer,

We made all of the changes as requested.

1- The FT-IR spectra in present form cannot be evaluated, because the bands are not clear; the differences are not well defined. Please interpret the results according to the literature. The X and Y axis titles are missing.

Thank you for your comment, we included as your request.

22. Please move the sentences “The SEM images were obtained using an Axio Imager m2m - Zeiss microscope.” and “The obtained images were increased by 20x. Image analysis was performed using Axiovision SE64 software.” in materials and methods section.

Thank you for your comment, we modified as your request.

Please, 

Find attached the certificate of the English Language review.

Sincerely,

Carlos Fernando Mourão

Reviewer 3 Report

I have carefully read the revised paper and believe that it is somewhat better than the previous submission.  The critiques addressed by the reviewer were answered for the most part satisfactorily.  The authors of the paper have clearly done a lot of work, and the results obtained by them are quite interesting. It reads better now.

Author Response

The authors sent the article for a Professional Native in the English language. 

Find attached the letter.

Sincerely,

Carlos Fernando Mourão 
